# The Impact of Music on Stress Biomarkers: Protocol of a Substudy of the Cluster-Randomized Controlled Trial Music Interventions for Dementia and Depression in ELderly Care (MIDDEL)

**DOI:** 10.3390/brainsci12040485

**Published:** 2022-04-08

**Authors:** Naomi L. Rasing, Sarah I. M. Janus, Gunter Kreutz, Vigdis Sveinsdottir, Christian Gold, Urs M. Nater, Sytse U. Zuidema

**Affiliations:** 1Department of General Practice and Elderly Care Medicine, University of Groningen, University Medical Center Groningen, 9700 AD Groningen, The Netherlands; s.i.m.janus@umcg.nl (S.I.M.J.); s.u.zuidema@umcg.nl (S.U.Z.); 2Department of Music, Speech and Music Lab, Carl von Ossietzky University Oldenburg, Ammerländer Heerstraße 114-118, 26129 Oldenburg, Germany; gunter.kreutz@uni-oldenburg.de; 3NORCE Norwegian Research Centre AS, Nygårdsgaten 112, 5008 Bergen, Norway; visv@norceresearch.no (V.S.); ch.go@norceresearch.no (C.G.); 4Department of Clinical and Health Psychology, University of Vienna, Liebiggasse 5, 1010 Vienna, Austria; urs.nater@univie.ac.at

**Keywords:** alpha-amylase, cortisol, music therapy, music-based therapies and interventions, saliva, hair, biomarkers, dementia, depression, elderly care

## Abstract

Recently, a large cluster-randomized controlled trial was designed—Music Interventions for Dementia and Depression in ELderly care (MIDDEL)—to assess the effectiveness of music interventions on depression in care home residents with dementia (ClinicalTrials.gov NCT03496675). To understand the pathophysiological mechanisms, we observed the effect of repeated music interventions on stress in this population since chronic stress was associated with depression and an increased risk for dementia. An exploratory study was designed to assess: (1) changes in hair cortisol concentrations as an indicator of longer-term stress; (2) whether baseline stress is a predictor of therapy outcome; (3) pre- and post-treatment effects on salivary α-amylase and cortisol response as an indicator of immediate stress in 180–200 care home residents with dementia and depressive symptoms who partake in the MIDDEL trial. Insights into mediatory effects of stress to explain the effect of music interventions will be gained. Hair cortisol concentrations were assessed at baseline and at 3, 6, and 12 months along with the Perceived Stress Scale. Salivary α-amylase and cortisol concentrations were assessed at 1, 3, and 6 months. Saliva was collected just before a session and 15 and 60 min after a session, along with a stress Visual Analogue Scale.

## 1. Introduction

Recently, a large cluster-randomized controlled trial was designed—Music Interventions for Dementia and Depression in ELderly care (MIDDEL)—from which the effectiveness of music interventions on depression in care home residents with dementia is assessed (ClinicalTrials.gov registration NCT03496675) [1]. Care home units (CHUs) and their residents are randomly allocated to one of four arms: group music therapy (GMT), recreational choir singing (RCS), the combination of both music interventions, or the control group receiving care as usual. Each intervention is offered twice a week for three months and once a week during the following three months. The primary outcome is depressive symptoms at 6 months, measured with the Montgomery–Åsberg Depression Rating Scale (MADRS) [2]. The RCT design, large sample size, and international collaborations within the MIDDEL trial (trial registration number NCT03496675) offer an excellent opportunity to implement this exploratory biomarker add-on study.

Besides investigating the effect of music on depressive symptoms, it is relevant to look at the effect of repeated music interventions on stress in care home residents with depressive symptoms, especially since chronic stress has been associated with depression and an increased risk for dementia. In the context of disease, stress—which is usually adaptive—can have negative effects on health and quality of life when prolonged [3,4]. Due to the body’s incapacity to cope with ongoing stress and the related physiological processes, so-called allostatic load occurs, which ultimately leads to illness [5]. In this sense, dementia is not caused by stress, but its manifestation might be facilitated by ongoing stress—and development might be accelerated (e.g., in care home residents) [6,7,8]. A recent review highlights the intricate association between depression and dementia and the accelerating and worsening effect that depression can have on the development of dementia. The authors attribute long-term dysregulations in the immune system, which can arise from chronic stress, as a possible underlying factor associated with both depression and dementia developments [9]. Similarly, Linnemann et al. (2020) described that stress, expressed in elevated cortisol levels and increased hypothalamus–pituitary–adrenal (HPA) axis activity, is one of the possible pathways connecting depression and dementia. Elevated cortisol levels were observed in 70% of depressed patients and also in people with Alzheimer’s, as shown in the review [10]. Chronic stress can elicit depression, yet, it is still unclear which neural pathways contribute to chronic stress leading to depression [11,12,13]. In a meta-analysis by de Witte et al. (2019) [14], music interventions were found to have a significant effect on the reduction in both physiological and psychological stress.

The response of the different components of stress-response systems can be measured using various biomarkers, such as cortisol and α-amylase [15]. Cortisol is a marker of the HPA axis [6,16]. Cortisol and α-amylase levels show an opposite rhythm throughout the day. Cortisol decreases during the day and increases at night; for α-amylase, vice versa [17]. Cortisol shows a peak approximately half an hour after awakening, whereas α-amylase typically declines at that time [17]. Cortisol concentrations then decrease throughout the day, whereas α-amylase increases and shows a peak late in the afternoon or in the evening [18]. Cortisol levels may fluctuate due to pulsatile secretion, the diurnal rhythm, and acute stress [19].

α-amylase is a marker for the autonomic nervous system (ANS) that can reflect stress-related changes [16,20]. The HPA axis and cortisol as markers of stress are well known and widely used. However, the importance and role of the ANS axis are becoming increasingly prevalent, with the salivary enzyme α-amylase being one of its reliable markers [5]. The activity of α-amylase increases in response to stress and can reflect the activity of the ANS and psychological and behavioral stress [4,5]. Indeed, previous studies found that psychological stress may affect sAA secretion, as it reflects ANS activity [21,22,23]. sAA was examined in behavioral medicine in different samples to identify patterns of dysregulations of the ANS (e.g., due to being exposed to chronic stress), gain insight into the application of sAA assessment to differentiate between treatment and control groups, and assess the effectiveness of various (stress-reducing) interventions [5]. Cortisol is commonly elevated in the elderly [8]. Elevated cortisol levels have been associated with poorer cognitive abilities, a higher risk of cognitive decline, and faster cognitive decline [7]. Several hair cortisol (HC) studies showed that chronic stress has a detrimental impact on daily functioning, engagement in social activities, and wellbeing/quality of life [24]. However, in a feasibility study, HC levels did not seem to be affected by medical conditions, cognitive dysfunction, psychotropic drug use, or behavioral problems [25] (p81). Cortisol levels may also be higher in people with depression [19,26]. To illustrate, Booij et al. [15] found higher SC and salivary α-amylase (SAA) levels, a steeper daily cortisol slope and a larger SAA over SC ratio in depressed than in non-depressed adult participants.

Cortisol and SAA were assessed in multiple psychotherapeutic intervention studies [6]. Non-pharmacological interventions in people with dementia may have a lowering effect on cortisol levels [7]. The two most commonly investigated pathways of these stress hormones are the use of pre-intervention cortisol concentrations as a treatment response predictor and the use of cortisol and α-amylase as an outcome measure to assess changes over time during the intervention period [6]. Interventions that aim to decrease HPA axis dysregulation may strengthen protective factors for the elderly. For example, such interventions may improve emotional regulation, offer social support, and reinforce resilience [8].

Biomarkers are increasingly used to investigate the potential stress-reducing effect of listening to music and music-making [16,26,27,28,29]. However, so far, relatively little research has been conducted to assess the effect of music-making on biomarkers, such as cortisol levels, in people with dementia and/or depressive symptoms.

The primary objective of this study was to assess the effect of repeated music interventions on chronic stress in care home residents with dementia and depressive symptoms. The differential effect of RCS, GMT, and RCS plus GMT was compared to care as usual. Secondary objectives were to assess whether the baseline level of chronic stress, as indicated by HC, is a predictor of therapy outcome (reduction in depressive symptoms) and to assess the immediate effect of music interventions on stress. As high depression scores correlate with higher HC concentrations, we hypothesized that participating in a music intervention would result in lower HC concentrations, with greater changes in the intervention groups than in the control group. The design of this study meets requirements to tackle frequent limitations in studies of HPA axis and/or ANS activity in music interventions in people with dementia, given the intervention duration, repeated assessments, control group for comparison, assessing both immediate and longer-term intervention effects [30].

## 2. Materials and Methods

### 2.1. Design

This study is a part of the large, multinational cluster-randomized controlled trial Music Interventions for Dementia and Depression in ELderly care (MIDDEL) (ClinicalTrials.gov registration NCT03496675) [1]. In the present exploratory substudy, participants from all four arms (GMT, RCS, GMT plus RCS, or control group) of the original trial from Norway, Germany, and the Netherlands were included (Figure 1). For the part of the substudy addressing immediate effects using saliva samples, only three arms (GMT, RCS, control group) were included. This limits participant burden and prevents uncertainty about which music intervention may have caused a certain effect in the combination group receiving both GMT and RCS. The primary outcome of the main trial was depressive symptoms at 6 months, measured with the MADRS [2]. For additional information about the main trial, study design, and procedures, see Gold et al., 2019 [1].

Chronic stress can be assessed objectively from cortisol in hair samples, as hair cortisol (HC) provides insight into cortisol levels over a longer period of time [4,25]. Sampling HC is an objective, easy, non-invasive, and affordable way of assessing chronic stress in people with dementia [19,24,25]. HC represents the longer-term retrospective activity of the HPA axis, which, in turn, is associated with chronic stress [31].

Measuring cortisol and α-amylase in the saliva is a non-invasive, easy, and reliable way to assess stress on health outcomes [3,4,5]. Assessment of stress in SC concerns real-time data, and especially in times of stress, there are significant correlations between cortisol concentrations in saliva and blood [4]. The digestive enzyme α-amylase can be measured from the same saliva sample to assess stress [5]. To date, SAA has only been assessed in a few clinical samples [5]. Hence, by examining both biomarkers—which can conveniently be analyzed with the same sample—a more complete picture of immediate physiological stress can be obtained [32].

### 2.2. Participants

Approximately 180–200 care home residents from 20 CHUs participating in the MIDDEL trial were included in this substudy (Figure 1). Care home residents were included if they were aged 65 years or older; resident (full-time, 24 h/day) at a participating CHU; had dementia as indicated by a Clinical Dementia Rating (CDR) score of 0.5 to 3 and a Mini-Mental State Examination (MMSE) score of 26 or less; had at least mild depressive symptoms, as indicated by the MADRS score of at least 8; had a clinical diagnosis of dementia according to the International Classification of Diseases and Related Health Problems 10 (ICD-10) research criteria; and provided written informed consent (IC) (optionally by proxy). Care home residents in different dementia stages were included. In line with the main MIDDEL trial protocol [1], where possible, the MADRS assessment was based on an interview with the participant. All relevant clues and information from other sources were used as a basis for the rating if the assessor was unable to obtain definite answers. Most participants were unable to report on themselves; hence, proxy informants were asked to report perceived stress on the Visual Analogue Scale.

The exclusion criteria were those diagnosed with Parkinson’s disease or schizophrenia, severe hearing impairment, being in short-term care, and not being able to sit down during (at least part of) the session. For saliva sample collection, additional exclusion criteria were having swallowing difficulties and being in the intervention group receiving both GMT and RCS. For hair sample collection, the additional exclusion criteria were those who did not have sufficient hair. In order to eliminate the risk of ingestion, residents with swallowing difficulties, indicated by physician and nurse reports, were excluded. Saliva samples were not collected in residents judged by care staff to be unable to follow the procedure, which may commonly be the case in residents with severe dementia (CDR = 3).

### 2.3. Randomization

Participants that were part of the MIDDEL trial were therefore randomized with their CHU to one of four groups: GMT, RCS, GMT plus RCS, or the control group. Randomization took place at the central study office after the completion of baseline assessments, using a computer-generated randomization list with a block size of four.

### 2.4. Intervention

A detailed description of the music interventions GMT and RCS were provided in the protocol article of the MIDDEL trial [1]. Care home units (CHUs) and their residents were randomly allocated to one of four arms: group music therapy (GMT), recreational choir singing (RCS), the combination of both music interventions, or the control group receiving care as usual. Each intervention was offered twice a week for three months and once a week during the following three months. Every session lasted approximately 45 min. GMT was provided in small groups (e.g., around five participants) by a qualified music therapist and included singing, improvising on instruments, and movement to music; RCS was provided in larger groups (e.g., with all residents of the unit) by a person with choir leading skills, and focused on singing. All units continued with standard care as locally available. The schedule, duration, and dose of intervention sessions are described in Figure 1.

### 2.5. Control Condition

The book reading served as a control condition for residents in the standard care group at T1 (within 1 month after completion of baseline assessment) and at T3 (3 months) and T6 (6 months). The control condition was similar to the music sessions in terms of group size, duration, and time of assessment. Hence, every book reading session took place around 15:00 h in the afternoon. All participating residents from the CHU participated in the book reading session. This session lasted 45 min. The book reading session was organized and supervised by either a volunteer, research assistant, research nurse, or care staff. This can also be the responsibility of one of the persons collecting samples before and after the session. The book reading session has an interactive character and functions as a social activity. A range of reading activities can be selected, such as reading local news stories, short stories, talking about headlines, and flipping through a magazine with pictures. Residents may read themselves or with the guidance of the supervisor if necessary. To illustrate, residents can read the paper and talk about headlines, whereas residents unable to read can flip through a magazine with pictures.

### 2.6. Outcome Measures

#### 2.6.1. Co-Primary Study Parameters: Hair Cortisol

Repeated assessment of HC is needed to assess the effect of repeated music interventions on chronic stress. HC provides insight into the amount of accumulated cortisol over a certain period (e.g., 1 or 3 months). Hair samples were collected at baseline (T0), at 3 months (T3), at 6 months (T6), and after 12 months (T12).

#### 2.6.2. Co-Primary Study Parameters: Perceived Stress Scale

Information from HC was complemented with subjectively experienced stress using the 10-item Perceived Stress Scale (PSS-10) [33]. This is a reliable and valid questionnaire that can be filled out by the resident or by proxy [34]. Stress experienced in the past month and feeling in control in stressful situations was assessed. The original questionnaire contains 14 items; however, the use of the 10-item version is recommended [35]. A higher sum score indicates higher perceived stress. Chronic stress was assessed with the PSS-10 at the same time as the hair samples were collected.

#### 2.6.3. Secondary Study Parameters: Salivary Cortisol and Alpha-Amylase

SC can provide more insight into how well the endocrine stress system responds to an immediate intervention because HC does not provide insight into how well the HPA axis functions. As residents with depressive symptoms were expected to have steeper cortisol slopes, we hypothesized that participants have a higher SC before the start of the music session, and participating in the music intervention attenuates the SC. Samples were collected on days a music or book reading session took place, between 12.00 and 17.00 h, preferably at 15.00 h, to minimize any confounding effects of the diurnal hormonal rhythm. In order to assess the immediate effects of music interventions on stress, saliva samples were collected to assess SC and SAA levels just before, 15 min after, and 60 min after the music/book reading session. Multiple measurements of SC and SAA per resident were required because of variations in levels over time within residents [15]. The assessment took place at T1 (within 1 month after the music interventions started or within 1 month after completion of the baseline assessment in the control group), T3 (3 months), and T6 (6 months). The days were selected according to logistic convenience (the first session that took around 15.00 h around T1, T3, and T6 were selected).

#### 2.6.4. Secondary Study Parameters: Proxy- or Self-Reported Experienced Stress

On days when saliva samples are collected, residents or a proxy reported any experienced stress using a Stress Visual Analogue Scale (VAS). The VAS is a vertical line of 100 mm with endpoints labeled ‘The least stress you can imagine’ (0) and ‘The most stress you can imagine’ (100). Residents or their proxies were asked to indicate on this line how stressed they felt at that exact moment. The scale thereby provides a single score between 0 and 100, representing subjective stress [36]. The VAS was filled out at T1, T3, and T6. Similar to the collection of saliva samples, the VAS was filled out—by proxy if necessary—just before, 15 min, and 60 min after the music or book reading session.

### 2.7. Data Collection Procedures

Biomarkers were collected throughout the trial to assess the immediate and longer-term effects of repeated music interventions on chronic and immediate stress in care home residents with dementia and depressive symptoms (Figure 2). In Figure 2, the stress assessments of the substudy are presented within the assessment schedule of the MIDDEL trial. Hair samples were collected in all groups at T0, T3, T6, and T12 to assess physiological changes in chronic stress. Saliva samples were collected in the RCS, the GMT, and the control group at T1, T3, and T6. Saliva samples were collected just before, 15 min, and 1 h after the intervention to assess the immediate intervention effects. The group GMT plus RCS were only included to assess the longer-term effect of repeated music intervention on (chronic) stress.

#### 2.7.1. Hair Sample Collection

In order to collect hair samples, research nurses cut hair samples of at least 3 to 4 cm long in all eligible residents at T0, T3, T6, and T12. Several thin hair strands were cut as close as possible to the scalp from the posterior vertex region of the head. A short questionnaire was filled out to assess hair characteristics. The University of Vienna developed detailed Standard Operating Procedures (SOPs) to ensure data collection takes place in the same manner in the three participating countries.

#### 2.7.2. Saliva Sample Collection

Residents were asked to collect the saliva using longer-length Salivettes (Salivettes, Sarstedt, Sevelen, Switzerland) and small cotton rolls originally devised for infants and young children, and the swab could be held by the research nurse, who was assisting with the sample collection. Although the use of Salivettes is not optimal to assess SAA, after careful consideration, we choose the use of longer-length Salivettes (SalivaBio Children’s Swab method) over the use of a passive drooling technique, as this was expected to be safe (eliminate the risk of ingestion), feasible, easy to learn and perform, and clean [20]. Hereby, we also took into account our expectation that a proportion of participating care home residents with dementia and depressive symptoms have difficulty understanding instructions and experience dry mouth due to age and medication use. Care staff received instructions prior to the day of sample collection, asking them to help residents—where possible—avoid physical exercise and chewing gum (24 h); consuming caffeinated and alcoholic beverages, juice, and tobacco (18 h); psychotropic drugs (6 h); brushing teeth; and eating or drinking (1 h) before the collection of the saliva sample. A short questionnaire was filled out by the proxy (care staff) to assess adherence to instructions prior to sample collection. Prior to the first collection of saliva samples (T1), residents practiced the saliva collection procedure when necessary to become familiar with the process.

There are different procedures to collect the saliva sample with the Salivette. It is recommended to keep the technique (stimulated or unstimulated) and location in the mouth similar across the study and residents because these affect the type of glands that produce the α-amylase that are activated and the amount of amylase that is secreted [20]. Saliva composition can differ significantly between stimulated and unstimulated saliva collected samples [37]. Several studies applied saliva sample collection methods in care home residents with dementia. Both unstimulated and stimulated saliva sample collection were applied previously in dementia research [38,39,40,41,42,43,44]. In some studies, specific saliva collection methods are not described in detail [45,46,47]. Mixed methods were applied when unstimulated collection did not lead to sufficient saliva to collect [37]. The current literature does not favor one method over the other. Hence, in the specific population of the current study, feasibility was the main selection criterion. The procedures to collect saliva samples in participating residents were carefully considered and pilot tested to assess which method is more appropriate. The saliva collection protocol applied in the current study is similar to methods applied by Pu et al. (2020), aiming at unstimulated saliva collection [42]. Where needed, mixed methods were applied. Residents were asked to either keep the roll in their mouth for approximately 2 min without chewing (thereby collecting unstimulated saliva) or, if needed, to softly chew on the roll for approximately 1 to 2 min (thereby collecting stimulated saliva). The roll was placed into a small tube and stored at −20 °C before analysis took place. Where possible, all samples of the same resident were assayed in the same batch, and SAA was assessed on the same day as SC.

### 2.8. Biochemical Analyses

Biochemical analyses of hair and saliva samples were performed at the biochemical laboratory of the University of Vienna. The University of Vienna conducted the analyses of biomarkers, α-amylase and cortisol from saliva samples and cortisol from hair samples. For determination of HC concentration in hair samples, the first scalp-near 3 cm segment was used, which is thought to reflect the cumulative cortisol secretion of the past 3 months [48]. Hair wash and cortisol extraction procedures were based on a previously established laboratory protocol [49]. In brief, hair samples were washed twice for 3 min using 3 mL isopropanol. For cortisol extraction, 10 ± 0.5 mg (alternatively, 7.5 ± 0.5 mg, dependent on the sufficient amount of hair provided across the participants’ sample) of whole, finely cut hair was incubated in 1.8 mL methanol for 18 h at room temperature. After incubation, 1.6 mL was transferred to another glass vial. Then, 1.6 mL of the supernatant was evaporated at 50 °C until samples were completely dried. Finally, the samples were re-suspended with 225 µL ultra-pure water and vortexed for 20 s. For cortisol determination, commercially available cortisol luminescence immunoassay was used (LIA, IBL, Hamburg, Germany). Saliva samples were stored at −20 °C until analysis. Free cortisol concentration in saliva was determined by using a commercial luminescence immunosorbent assay (LUM; IBL, Hamburg, Germany). SAA activity was measured using an enzymatic colorimetric test [50] and reagents obtained from DiaSys Diagnostic Systems (Holzheim, Germany). Saliva was diluted at 1:625 using 0.9% saline solution. The reagents contain the enzyme α-amylase in a specified amount and alpha-glucosidase, which converts the substrate ethyliden nitrophenylto p-nitrophenol. The rate of formation of p-nitrophenol was directly proportional to the samples’ amylase activity and was detected using an absorbance reader at 405 nm (Biotek Synergy HTX, Biotek Instruments, Winooski, VA, USA). Inter- and intra-assay coefficients of variation were calculated to assess precision of the analytic method and were expected to be below 10% for all assays. All samples of one subject were analyzed in the same batch, if possible, to reduce error in variance. Saliva samples with an insufficient amount of saliva to analyze were excluded from statistical analysis.

### 2.9. Sample Size and Power

Through convenience sampling, a subsample of residents from three countries (the Netherlands, Germany, and Norway) from the MIDDEL trial was asked to participate in this substudy. The aim was to include 50 care home residents in every group (GMT, RCS, GMT plus RCS, control group at baseline). The Netherlands and Germany aimed to include 8 CHUs/80 participants each, and Norway aimed to include 4 CHUs/40 residents. Attrition of residents was expected to occur due to mortality, moving to another care home, or refraining from participation in the substudy. The mean length of stay in a care home in the Netherlands is 8 to 12 months. A mortality rate of 20% within the intervention period of 6 months (primary outcome at 6 months) was taken into account with respect to the sample size. Mortality is lower in care home residents with mild to moderately severe dementia (compared to severe and very severe). The power for this exploratory substudy with this given sample size was difficult to determine because there are no previous studies or agreed minimal clinically important difference for this combination of intervention, population, and outcome. As in the MIDDEL protocol [1], the main effects in this substudy were tested as GMT versus no GMT and RCS versus no RCS so that all participants with available data could be used in each test. Thus, after a dropout of 20% until the primary endpoint of 6 months, as assumed in the MIDDEL protocol, we expected to have 160 participants in total. Compared to the MADRS, we expected the cluster effect in these biological indicators to be smaller, with an ICC ≤ 0.01. The average cluster size after 20% dropout will be eight. Because there are two main comparisons, tests were Bonferroni-corrected from 5% to 2.5%, two-sided. We used R to determine the effect size for which 80% power is reached with a two-sample *t*-test with 160 participants in total, adjusted for clustering, with a two-sided 2.5% significance level, and found that the effect size that can be reliably determined with this substudy is d = 0.51, i.e., a medium effect size [51]. This seems reasonable compared to previous studies of music therapy [14,52]. The actual power may be higher in a longitudinal model if time points are correlated.

### 2.10. Data Analysis

#### 2.10.1. Co-Primary Study Parameters

Stress (HC concentrations and perceived stress measured with the PSS-10) after 3 and 6 months was modeled to account for differences in therapy (RCS compared to no RCS, GMT compared to no GMT, additional effects of GMT + RCS combined, and baseline differences in stress (HC concentrations and perceived stress measured with the PSS-10) (T0).

#### 2.10.2. Secondary Study Parameters

In the final model in which MADRS scores at the endpoints (3 and 6 months) are predicted by the therapy group (Gold et al., 2019), baseline stress was added to examine whether chronic stress (HC concentrations; perceived stress) predicts the treatment effect. Secondary study parameters were added to the final model. A description of the effect of amylase response (expected to change at 15 min), cortisol response (expected to change at 60 min), and experienced stress (assessed with the VAS) were provided. Correlation of stress biomarker response (SC and SAA levels at T3 and T6, pre-post treatment) and change in MADRS score (T3–T0 and T6–T0) were examined. The design was able to account for between-group variation (including differential responses in three groups) and within-group variation (changes in participants between 3 months, 6 months, and baseline).

### 2.11. Statistical Analyses

Data were analyzed using SPSS Statistics version 26. Immediate and chronic stress were assessed with psychological subjective indicators of stress and physiological markers of stress. Descriptive statistics (mean, standard deviation, range) were calculated for demographics, the PSS, and VAS scores for all time points. Power calculations were computed. In addition, we reported effect size estimates. The distribution was assessed, and if needed, data were log-transformed to be normally distributed accordingly, especially since cortisol and SAA data may have a positively skewed distribution.

The effect of repeated music interventions on chronic stress was measured with the biomarker HC. HC concentrations were reported in pg/mg. Concentrations of HC at baseline were assessed as a predictor of treatment response. In order to assess the association between the HC concentration and perceived stress (PSS-10), correlations were computed. Changes in HC concentrations over time and perceived stress (PSS-10) were assessed. The levels of stress after 3 and 6 months were modeled to account for differences in therapy: RCS compared to control (usual care or book reading), GMT to control, GMT + RCS, and baseline differences in stress (T0). In the final model, in which MADRS scores at the endpoints (3 and 6 months) were predicted by the therapy group, baseline stress was added to observe whether there was a change in treatment effect. The analyses included a correlation between stress reduction (at T3, pre-post treatment) and change in MADRS score (T6–T3–T0).

SC and SAA were concomitantly measured and were used as outcome measures to assess whether changes in activity in the HPA axis and the ANS axis occurred pre- and post-intervention. Pre- and post-sampling weight of saliva samples were measured. Levels of SC were reported in nmol/L and levels of SAA in U/mL. Adherence to instructions prior to the collection of the saliva sample was assessed. A description of the effect of SAA response (expected to change at 15 min) and SC response (expected to change at 60 min) was provided. The design is able to account for between-group variation (including differential responses in four groups) and within-group variation (changes in participants between 3months, 6 months, and baseline). Exploratory subgroup analyses and moderator analyses were performed.

## 3. Discussion

To the best of our knowledge, this is the first exploratory study assessing the effect of repeated music interventions (GMT and RCS) on chronic and acute stress using samples of HC, SC, and SAA in care home residents with dementia and depressive symptoms in the context of the large, multinational cluster-randomized controlled MIDDEL trial.

The collection of SC, SAA, and HC in care home residents with dementia might come with several specific challenges, such as apraxia, reduced saliva production because of aging, comorbidity, anticholinergic medication, and neurological/cognitive dysfunction in general. In the design and execution of the current study, we considered recommendations of previous studies. Firstly, levels of SC and SAA change over the lifespan [32]. An important consideration is the finding that the use of drugs, such as antidepressants and analgesics, can lead to insufficient saliva flow [32]. Several studies found that specific types of drugs might affect basal SC concentrations. Other aspects that can influence SC and SAA are food and consumption, alcohol and tobacco use, sleep, and exercise. A recent study found that physical frailty may also influence SC, although people with severe dementia were excluded [53]. Important potential confounders are: age, sex, somatic conditions (both somatic and acute) and related regular medication, smoking status, food and drink consumption (in the past hour), alcohol consumption (in the past 24 h), physical activity (in the last hour), and sleep (awakening time and napping in the past hour) [32]. Confounders associated with the outcomes (*p* < 0.05) were statistically controlled for.

Secondly, when collecting saliva samples, it is recommended to consider the sampling technique. The amount of saliva and SAA can be affected by the way samples are collected: stimulated samples by chewing on a small roll of cotton or unstimulated by passively drooling into a tube [5,53]. As the salivary flow rate can be limited in the target group because of aging and medication use, the use of a cotton roll (Salivette) was preferred. Another recommendation to better understand interactions is to include different types of biomarkers and multiple data collection time-points [15,16]. Ideally, cortisol concentrations are measured with a cortisol awakening response (CAR), with measurements up to seven times a day for two consecutive days. However, this is not the main question of this exploratory substudy; therefore, a more minimalistic design with three measurements on day one was chosen. Similar to other studies assessing changes in biomarkers in response to music (therapy) [46,54], in the current study, SC and SAA levels were assessed with multiple measurements, pre- and post-session. This will provide insight into changes within and between residents as a result of the music intervention.

Lastly, when designing a study assessing stress-related biomarkers, the timing of assessment, sampling periods, and consideration of psychobiological interactions should be taken into account [27]. Furthermore, an explanation of the bodily mechanisms of the biomarkers under investigation should be provided, thereby underlining the significance of testing these specific biomarkers [28]. Describing how the hormone is produced, how its levels are increased or decreased, and what its impact is on the body and the immune system is crucial to be able to put results and its relevance into perspective [28]. Moreover, the types of music should be clearly defined to be able to attribute changes in the biomarkers measured to a specific aspect of the music intervention, such as mode of delivery. This is the case in the music interventions offered in the MIDDEL trial.

## 4. Conclusions

This is a sub-study of a large, multinational cluster-randomized trial investigating the effect of music interventions GMT and RCS in care home residents with dementia and depressive symptoms. In this trial, 100 CHUs from six different countries will participate, and 1000 care home residents are expected to be included. A subsample of residents from three countries (the Netherlands, Germany, and Norway) will be asked to participate in this substudy. The assessment of biomarkers cortisol and α-amylase in saliva and cortisol in hair will provide insight into the potential beneficial effects of GMT and RCS on stress in care home residents with dementia and depressive symptoms. The assessment of cortisol and α-amylase may contribute to unraveling the treatment effects of psychosocial interventions in general and music interventions specifically. Ultimately, this exploratory add-on study contributes to underpinning physiological processes associated with stress responses in care home residents undergoing a musical treatment and adds another dimension or perspective to the findings of treatment effects and outcome measures of the MIDDEL trial. Repeated music interventions may be able to reduce stress in care home residents with dementia and depressive symptoms and thus, to some extent, affect the course of the disease. If care home residents benefit from music interventions clinically, this knowledge can also be used for prevention.

In conclusion, this study is adequately designed, sufficiently powered, and feasible to determine relevant effects on biological and psychological measures of chronic stress. As the first study of its kind, it will contribute new knowledge about the mechanisms of music interventions for older care home residents with dementia and depression.

## Figures and Tables

**Figure 1 brainsci-12-00485-f001:**
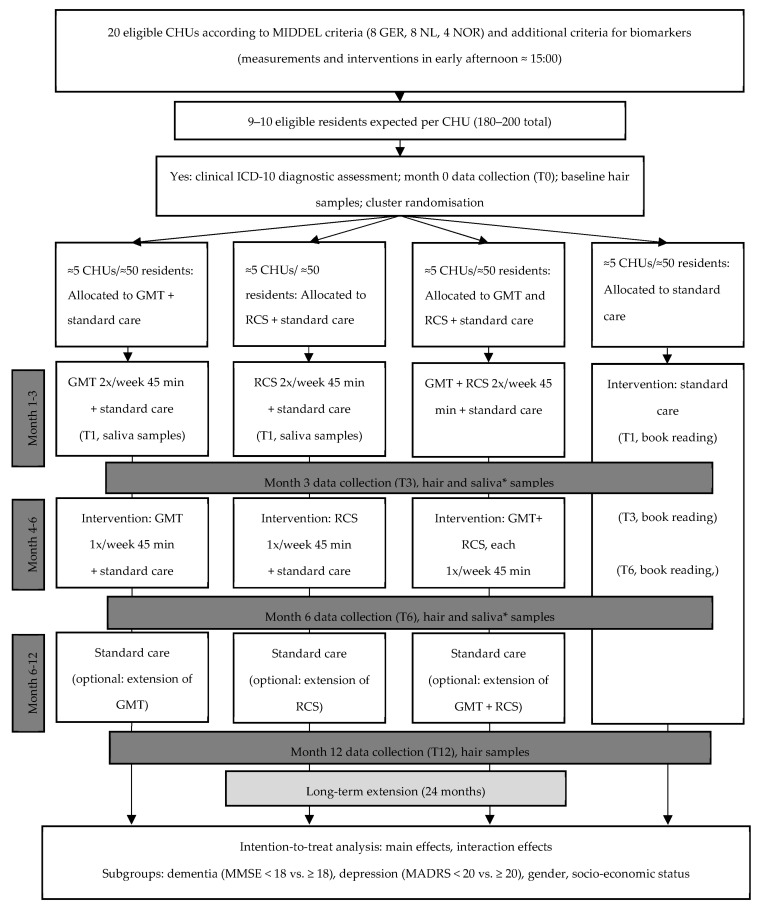
Flowchart illustrating the design of the substudy integrated within the Music Interventions for Dementia and Depression in ELderly care (MIDDEL) trial. * In the CHUs in the combination group (GMT + RCS), no saliva samples were collected.

**Figure 2 brainsci-12-00485-f002:**
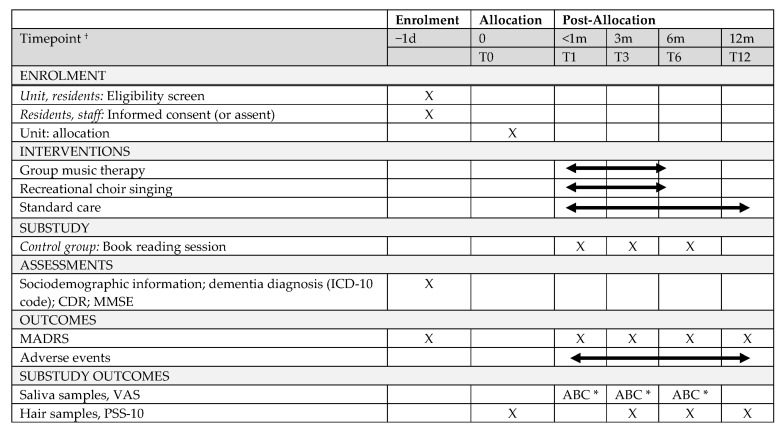
Schedule of enrolment, interventions, and assessments of the exploratory substudy within the MIDDEL-project. ^†^ T0: Before the music sessions start; T1: Within 1 month after the music interventions started or within 1 month after completion of the baseline assessment in the control group; T3: 3 months after the start of the music sessions; T6: 6 months after the start of the music sessions; T12: 12 months after the start of the music sessions. * A. Ten minutes before session; B. Fifteen minutes after session; C. Sixty minutes after session.

## Data Availability

The data presented in this study will be available on request from the corresponding author. This study protocol does not contain any data, as data is still to be collected during the trial.

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
