# Peer review of "The Impact of Music on Stress Biomarkers: Protocol of a Substudy of the Cluster-Randomized Controlled Trial Music Interventions for Dementia and Depression in ELderly Care (MIDDEL)"

_brainsci, 2022, doi:10.3390/brainsci12040485_

Round 1
Reviewer 1 Report
The paper described the protocol of a subsidy of the MIDDEL project aiming to explore the potential biological mechanism of music intervention for dementia and depression in older adults.
Generally, the article was well written straightforwardly. However, there are a couple of issues that need further clarification or modification:
- Improvement in the depressive symptoms was the primary outcome of the parent study. The current substudy aims to examine the change of perceived stress. However, the Introduction lacked literature on the potential link between depression and perceived stress among persons living with dementia (PLWD).
- The study will recruit PLWDs of different dementia stages. How will the study team collect data on MADRS and perceived stress if the PLWD is at an advanced stage? Are there any exclusion criteria in terms of PLWD’s verbal communication competency?
- Concerning collecting saliva samples, it would be better if the protocol specified which technique has been determined: stimulated or unstimulated? The paper stated that pilot tests would determine which method is more appropriate. However, the method should be determined based on the scientific rationale rather than the convenience or other factors unrelated to the hypothesis.
- In 2.11.2 (data analysis) Secondary study parameters: The measurement of SC and SAA should be modeled. The MADRS score is the primary score of the parent study but not the secondary study parameters. This paragraph should be tailored to the present substudy.
Minor comments:
Line 115: “trail” should be “trial”
Line 297: The paper only mentioned two methods for saliva sample collection. “most appropriate” should be “more appropriate.”
Outcome measures:
(1) Level of perceived stress should be considered as the co-primary study parameters
(2) Line 197: “saliva samples” cannot be referred to as “secondary study parameters.” It should be “saliva cortisol, saliva alpha-amylase, and self-reported (or proxy-reported) experienced stress.”
Line 373-375: The sentence is incomplete.
Discussion: Subtitle 3.1 is not necessary.
Author Response
Response to Reviewer 1
The paper described the protocol of a subsidy of the MIDDEL project aiming to explore the potential biological mechanism of music intervention for dementia and depression in older adults. Generally, the article was well written straightforwardly. However, there are a couple of issues that need further clarification or modification.
We thank the reviewer for the helpful suggestions to further improve the manuscript. Below we provide a point-by-point response to every issue.
Point 1:
Improvement in the depressive symptoms was the primary outcome of the parent study. The current substudy aims to examine the change of perceived stress. However, the Introduction lacked literature on the potential link between depression and perceived stress among persons living with dementia (PLWD).
Response 1: Thank you for mentioning this. We added findings from previous research to point out this potential link to the introduction. It now says:
“A recent review highlights the intricate association between depression and dementia, the accelerating and worsening effect that depression can have on the development of dementia. The authors attribute long-term dysregulations in the immune system, which can arise from chronic stress, as a possible underlying factor associated with both depression and dementia developments (Hayley et al., 2021). Similarly, Linnemann et al (2020) describe that chronic stress, expressed in elevated cortisol levels and increased HPA-axis activity, is one of the possible pathways connecting depression and dementia. Elevated cortisol levels have been observed in 70% of depressed patients, but also in people with Alzheimer's disease, the review shows. Chronic stress can elicite depression, yet, it is still unclear which neural pathways contribute to chronic stress leading to depression (Aznar et al., 2011; Pariante et al., 2008; Zheng et al., 2022). In a meta-analysis of de Witte et al. (2019) music interventions have been found to have a significant effect on reduction of both physiological and psychological stress.
Point 2: The study will recruit PLWDs of different dementia stages. How will the study team collect data on MADRS and perceived stress if the PLWD is at an advanced stage? Are there any exclusion criteria in terms of PLWD’s verbal communication competency?
Response 2: Thank you for pointing this out. These issues are elaborated in the protocol article of the main MIDDEL trial. We now added more information about this in the current manuscript as well.
“Care home residents in different dementia stages will be included. In line with the main MIDDEL trial protocol, where possible, the MADRS assessment will be based on an interview with the participant. All relevant clues and information from other sources can be used as a basis for the rating, if the assessor is unable to get definite answers. Most participants are unable to report on themselves, hence, proxy informants will be asked to report perceived stress on the Visual Analogue Scale.”
We also added a description of exclusion criteria.
“Exclusion criteria are being diagnosed with Parkinson’s disease or schizophrenia, having severe hearing impairment, being in short-term care, and not being able to sit down during (at least part of) the session. For saliva sample collection additional exclusion criteria are: having swallowing difficulties, and being in the intervention group receiving both GMT and RCS. For hair sample collection the additional exclusion criteria is not having sufficient hair.”
Point 3: Concerning collecting saliva samples, it would be better if the protocol specified which technique has been determined: stimulated or unstimulated? The paper stated that pilot tests would determine which method is more appropriate. However, the method should be determined based on the scientific rationale rather than the convenience or other factors unrelated to the hypothesis.
Response 3: We agree with this suggestion and added information about saliva collection methods described in previous studies concerning saliva sample collection in elderly with dementia. However, information is scarce. We added the following:
“Saliva composition can differ significantly between stimulated and unstimulated saliva samples (Gomar-Vercher et al., 2018). Several studies applied saliva sample collection methods in care home residents with dementia. Both unstimulated and stimulated saliva sample collection have been applied previously in dementia research (Bourne et al., 2019; Emami et al., 2022; Kwan et al., 2016; Liu et al., 2017; Schaub et al., 2018; Pu et al., 2020; Woods et al., 2011). In some studies specific saliva collection methods are not described into detail (Chu et al., 2014; de la Rubia Orti et al., 2018; Theorell et al., 2021). Mixed methods have been applied when unstimulated collection did not lead to sufficient saliva to collect (Gomar-Vercher et al., 2018). Current literature does not favor one method over the other. Hence, in the specific population of the current study, feasibility is the main selection criterion. The procedures to collect saliva samples in participating residents was carefully considered and pilot tested to assess which method is more appropriate. The saliva collection protocol applied in the current study is similar to methods applied by Pu et al (2020), aiming at unstimulated saliva collection. Where needed, mixed methods will be applied (Gomar-Vercher et al., 2018).”
Point 4: In 2.11.2 (data analysis) Secondary study parameters: The measurement of SC and SAA should be modeled. The MADRS score is the primary score of the parent study but not the secondary study parameters. This paragraph should be tailored to the present substudy.
Response 4: Thank you for pointing this out. As this is an exploratory substudy, we are not able to appraise what effect sizes we can anticipate. Secondary study parameters will be added in the final model. We aimed to tailor paragraph 2.11.2 accordingly.
Minor comments:
- Line 115: “trail” should be “trial”
We corrected this error.
- Line 297: The paper only mentioned two methods for saliva sample collection. “most appropriate” should be “more appropriate.”
Thank you, we corrected this as suggested.
- Outcome measures:
- (1) Level of perceived stress should be considered as the coprimary study parameters.
We agree, chronic stress measured in hair cortisol and stress indicated by the Perceived Stress Scale are coprimary study parameters. In paragraph 2.5.1 hair cortisol is addressed, in paragraph 2.5.2 the Perceived Stress Scale is addressed. Paragraph headings are adjusted accordingly.
- (1) Level of perceived stress should be considered as the coprimary study parameters.
- (2) Line 197: “saliva samples” cannot be referred to as “secondary study” It should be “saliva cortisol, saliva alpha-amylase, and self-reported (or proxy-reported) experienced stress.”
We changed the previous phrase to “Secondary study parameters: salivary cortisol and salivary alpha-amylase” for paragraph 2.5.3. We changed the heading of paragraph 2.5.4 to “Secondary study parameters: proxy- or self-reported experienced stress”.
- Line 373-375: The sentence is incomplete.
We now completed the sentence.
“A description of effect of amylase response (expected to change at 15 minutes) and cortisol response (expected to change at 60 minutes) will be provided. Correlation of stress biomarker response (SC and SAA levels at T3 and T6, pre-post treatment) and change of MADRS score (T3 – T0 and T6 – T0) will be examined.” - Discussion: Subtitle 3.1 is not necessary.
As suggested we removed subtitle 3.1.
Reviewer 2 Report
This protocol paper was clearly written with plenty of methodological details and a sound plan for analysis as well as good understanding of constraints, issues that may arise etc. I didn't see anything specific that required clarification or improvement.
Author Response
Response to Reviewer 2
This protocol paper was clearly written with plenty of methodological details and a sound plan for analysis as well as good understanding of constraints, issues that may arise etc. I didn't see anything specific that required clarification or improvement.
We thank the reviewer for the positive evaluation of the manuscript.
Reviewer 3 Report
The work clearly establishes the hormonal pathway of cortisol, but does not detail the alpha-amylase pathway, it would be convenient to establish more clearly its use and its "reverse" function in relation to cortisol (Krahel et al. Mediators Inflamm. 2021 Nov 12; 2021:3639441. doi: 10.1155/2021/3639441).
It is necessary to include the exclusion criteria for the study.
Author Response
Response to Reviewer 3
We thank the reviewer for the helpful suggestions to further improve the manuscript. Below we provide a point-by-point response to every issue.
Point 1: The work clearly establishes the hormonal pathway of cortisol, but does not detail the alpha-amylase pathway, it would be convenient to establish more clearly its use and its "reverse" function in relation to cortisol (Krahel et al. Mediators Inflamm.2021 Nov 12; 2021:3639441. doi: 10.1155/2021/3639441).
Response 1: We agree with the reviewer that we could more clearly establish the alpha-amylase pathway. We tried to do so, using the suggested literature.
“Cortisol shows a peak approximately half an hour after awakening, whereas alpha-amylase typically declines at that time (Nater, 2007). Cortisol concentrations then decrease throughout the day, whereas alpha-amylase increases and shows a peak late in the afternoon or in the evening (Skoluda, 2017).”
“The HPA-axis and cortisol as markers of stress are well known and widely used. Yet, the importance and role of the ANS-axis is becoming increasingly prevalent, with the salivary enzyme alpha-amylase being one of its reliable markers [5]. The activity of α-amylase increases in response to stress and can reflect activity of the ANS and psychological and behavioral stress [4,5]. Indeed, previous studies found that psychological stress may affect sAA secretion, as it reflects ANS activity [21–23]. sAA has been examined in behavioral medicine in different samples to identify patterns of dysregulations of the ANS (e.g. due to being exposed to chronic stress); to gain insight into application of sAA assessment to differentiate between treatment and control groups; and to assess effectiveness of various (stress-reducing) interventions [5].”
Point 2: It is necessary to include the exclusion criteria for the study.
Response 2: We added the exclusion criteria. It now says:
“Exclusion criteria are being diagnosed with Parkinson’s disease or schizophrenia, having severe hearing impairment, being in short-term care, and not being able to sit down during (at least part of) the session. For saliva sample collection additional exclusion criteria are: having swallowing difficulties, and being in the intervention group receiving both GMT and RCS. For hair sample collection the additional exclusion criteria is not having sufficient hair.”